# Laboratory realization of relativistic pair-plasma beams

C. D. Arrowsmith [1] ✉, P. Simon [2,3], P. J. Bilbao [4], A. F. A. Bott [1], S. Burger[2], H. Chen [5], F. D. Cruz[4], T. Davenne[6], I. Efthymiopoulos[2], D. H. Froula [7], A. Goillot[2], J. T. Gudmundsson [8,9], D. Haberberger[7], J. W. D. Halliday [1], T. Hodge[1,10], B. T. Huffman[1], S. Iaquinta[1], F. Miniati[1], B. Reville [11], S. Sarkar [1], A. A. Schekochihin[1], L. O. Silva [4], R. Simpson[5], V. Stergiou[1,2,12], R. M. G. M. Trines [6], T. Vieu[11], N. Charitonidis [2], R. Bingham [6,13] & G. Gregori [1]

Relativistic electron-positron plasmas are ubiquitous in extreme astrophysical environments such as black-hole and neutron-star magnetospheres, where accretion-powered jets and pulsar winds are expected to be enriched with electron-positron pairs. Their role in the dynamics of such environments is in many cases believed to be fundamental, but their behavior differs significantly from typical electron-ion plasmas due to the matter-antimatter symmetry of the charged components. So far, our experimental inability to produce large yields of positrons in quasi-neutral beams has restricted the understanding of electron-positron pair plasmas to simple numerical and analytical studies, which are rather limited. We present the first experimental results confirming the generation of high-density, quasi-neutral, relativistic electron-positron pair beams using the 440 GeV/c beam at CERN's Super Proton Synchrotron (SPS) accelerator. Monte Carlo simulations agree well with the experimental data and show that the characteristic scales necessary for collective plasma behavior, such as the Debye length and the collisionless skin depth, are exceeded by the measured size of the produced pair beams. Our work opens up the possibility of directly probing the microphysics of pair plasmas beyond quasi-linear evolution into regimes that are challenging to simulate or measure via astronomical observations.

Relativistic electron-positron ($e^\pm$) pair plasmas are expected to be produced around black holes[1] and neutron stars[2]. In these environments, pair creation can occur due to intense, high-energy $\gamma$-ray fluxes (by the Breit-Wheeler process[3]) or when the electromagnetic fields are comparable to the Schwinger field: the critical field strength for vacuum breakdown ($E_c = 1.3 \times 10^{18}$ V/m, $B_c = 4.4 \times 10^9$ T)[4,5]. Because of the symmetry of the charged components, electron-positron pair plasmas should exhibit collective behavior that is significantly different from typical electron-ion plasmas[6]. Linear and non-linear wave

processes can be affected in both fluid and kinetic regimes because of the suppression of some wave modes. This is important in a variety of astrophysical settings, with recent attention focusing on fast radio burst generation and the stability of astrophysical pair beam jets[7–11]. However, producing sufficiently large yields and densities of $e^\pm$ pairs in the laboratory in order to directly probe the relevant plasma microphysics has been challenging. Presently, high flux laboratory sources of positrons include: (i) nuclear reactors[12], (ii) electron accelerators[13,14], and (iii) high-power lasers[15–21]. All these approaches involve pair

**Fig. 1 | Experimental setup.** Protons with 440 GeV/c momentum are extracted from the SPS ring with maximum intensity of $3 \times 10^{11}$ protons in a single bunch of duration 250 ps (1-$\sigma$), and transverse size $\sigma_r = 1$ mm. The transverse beam profile of the secondary beam is imaged using a 70 mm × 50 mm × 0.25 mm chromium-doped (Chromox) luminescence screen positioned 10 cm downstream of the target, and a blocker foil (50 µm aluminum) is used to minimize stray optical light. The Chromox screen is oriented at 45° to the beam path and viewed by a digital camera which has an exposure time of 24 ms to capture the entire scintillation of the screen. The 3.8 m standoff distance of the digital camera leads to image resolution of 50 µm, however the actual resolution is 100 µm due to the translucence of the

Chromox. At a distance 2 m downstream of the target, electrons, and positrons are separated from the secondary beam and spectrally resolved using a magnetic spectrometer comprised of an electromagnet and a pair of luminescence screens (200 mm × 50 mm × 1 mm) centered at a distance 240 mm off-axis. 20-cm thick bricks of concrete (not shown in the diagram) are placed at the entrance of the electromagnet, leaving a 40 mm-wide aperture. Concrete is also placed to block the target from the direct view of the cameras to minimize speckle background arising on the camera images from the impact of high-energy hadrons scattered around the experimental area.

production processes when sufficiently energetic $\gamma$ rays ($E_\gamma \geq 2m_e c^2 = 1.022$ MeV) interact with charged nuclei (so-called Trident and Bethe-Heitler processes[22]), with the highest cross-section in high-Z materials. In the coming decade, it is proposed to use magnetic chicanes at FACET-II (SLAC) to combine the accelerator's e⁺ and e⁻ beams into a quasi-neutral jet[23]. The next generation of ultra-intense lasers may also be able to produce pairs by achieving the Schwinger limit for vacuum breakdown[24–26]. Meanwhile, precision magnetic confinement techniques have been developed to trap low-temperature e± pair plasmas[27–29], and relativistic laser-produced plasmas[30–32]. However, despite significant efforts, none of these approaches have so far been able to produce the yields and densities of pairs needed to sustain collective modes in the plasma.

Here we present a novel approach for producing quasi-neutral e± jets in which a high-intensity, ultra-relativistic proton beam is converted into pairs via hadronic and electromagnetic cascades with 2–3 orders of magnitude higher yield than previously reported neutral beams[17,19]. We performed our experiment at the HiRadMat (High-Radiation to Materials) facility[33] in the accelerator complex at CERN, Geneva. The experimental setup is shown in Fig. 1. We performed detailed Monte-Carlo simulations using the standard computer code FLUKA[34–36], which uses a robustly bench-marked physics model to characterize the e± pair production as well as the other secondary beam components (hadrons and $\gamma$ rays). The predicted number of pairs produced with kinetic energy greater than 1 MeV is $N_\pm = \frac{1}{2}(N_{e^+} + N_{e^-}) = 1.5 \times 10^{13}$, with peak pair density $n_\pm = 1.6 \times 10^{12}$ cm⁻³, and the ratio of positrons to electrons is $N_{e^+}/N_{e^-} = 0.82$. Downstream of the target, the positron ratio is even higher ($N_{e^+}/N_{e^-} > 0.9$) as the discrepancy between the e⁻ and e⁺ spectra exists only for the lowest energy pairs, which preferentially escape the beam due to their higher divergence.

The large numbers of electron-positron pairs are generated using a single LHC-type bunch of $3 \times 10^{11}$ protons with momentum 440 GeV/c and duration (1-$\sigma$) of $\tau = 250$ ps. The protons are extracted to the facility from the Super Proton Synchrotron, irradiating a solid target composed of a low-Z material (graphite) and a high-Z converter (tantalum). The dominant process for producing electron-positron pairs is

the hadronization of quarks and gluons inside the graphite section of the target. This produces a shower of pions, kaons, and other hadrons on scales comparable to the nuclear interaction length in graphite[37]. A copious number of ultra-relativistic neutral pions are produced, which almost instantaneously undergo electromagnetic decay to produce a highly collimated flux of GeV-scale $\gamma$ rays. Electromagnetic cascades are then generated with the $\gamma$ rays producing pairs in the high-Z tantalum converter, which is much longer than its radiation length. Further pairs are created via subsequent bremsstrahlung of electrons and positrons (Bethe-Heitler process). Secondary $\gamma$ rays that do not convert into pairs can escape the target, along with a much smaller number of protons and other hadronic species (by orders of magnitude). The effect of hadronic beam components (such as $\pi^\pm$ pairs) on e± pair-plasma dynamics must be considered, but the effects are expected to be negligible due to the lower mobility and density of these species. The choice of target material length constitutes a compromise between the number of pairs produced and the emittance of the e± beam, with the thickness of graphite and tantalum chosen in the current setup to maximize the pair density ($n_\pm$) maintained over a 1 m length downstream of the target. Choosing thicker target materials can produce an even greater yield and density of e± at the immediate rear of the target[37]. Details of the beam characteristics of all the secondary components are provided in the Supplementary Information. FLUKA Monte-Carlo simulations play a critical role in interpreting the experimental measurements of the e± pairs. Where the pair density is highest (at the immediate rear of the target), it is not possible to directly measure certain properties of the pair population, such as the precise distribution function (energy/momentum spectrum).

Experiments are performed to measure the transverse beam profile and the e± energy spectra downstream of the target (see Fig. 1 for details), allowing validation of FLUKA simulations. In both measurements, pair fluence is measured using luminescence screens made of chromium-doped alumina-ceramic (Chromox)[38–40]. When ionizing particles or radiation are incident on the screen, the screen emits red visible light with a few ms decay time. For our conditions, the intensity of light is directly proportional to the energy deposited by ionizing

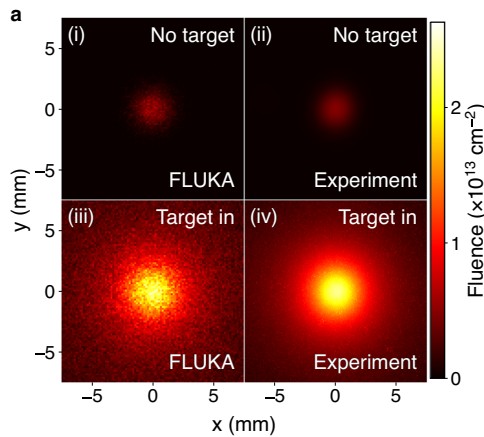
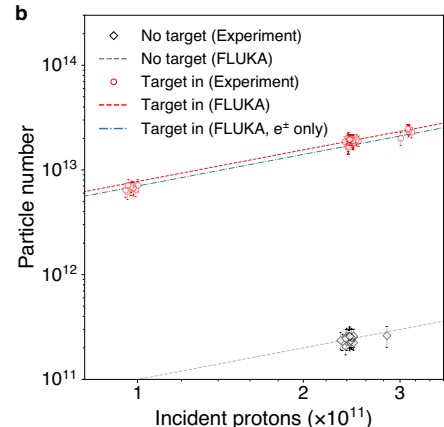

**Fig. 2 | Transverse beam profile imaged using a luminescence screen. a** Direct comparison of FLUKA Monte-Carlo simulations with raw image data obtained when the target is irradiated and the secondary beam is produced ('Target in'), versus when the target is removed, and only the primary proton beam irradiates the screen ('No target'). An absolute fluence calibration is obtained using the known density profile of the primary proton beam. **b** Integrated image intensity (total intensity) from 68 shots is converted to an absolute particle number, showing the case where the target is irradiated (red circles, 46 shots), and when it is removed (black diamonds, 22 shots). The error bars reflect the standard errors of the fitted parameters for each shot. FLUKA Monte-Carlo simulations of the predicted light yield are shown for both cases (black-dashed and red-dashed lines), showing good agreement with the experimental data. The blue dot-dashed line indicates the contribution from e$^{\pm}$ in the FLUKA simulation, highlighting that this is the dominant contribution to the enhanced signal.

particles (see Methods). Although it is not possible to distinguish which kind of secondary particle is the cause of observed luminescence, at relativistic energies (Lorentz factor, $\Gamma \gtrsim 2$) charged particles deposit energy in a 'minimum-ionizing' fashion, where an almost identical amount of energy is deposited by each particle (simulations confirming this are presented in the Supplementary Information). Given that the vast majority of secondary particles incident on the screen are relativistic, the observed brightness is thus assumed to be directly proportional to the fluence of incident particles. Therefore a secondary beam containing ~$10^{13}$ e$^{\pm}$ pairs is expected to produce a ~100 times larger luminescence intensity compared with a primary beam containing ~$10^{11}$ protons. Since the target is mounted onto a motorized stage, it can be entirely removed from the path of the primary proton beam, allowing the screen to be directly irradiated. By independently measuring the incident proton beam fluence for each shot using upstream current monitors, an absolute calibration of the particle fluence can be made.

A common source of background in experiments producing energetic electromagnetic cascades is a large number of scattered sub-MeV e$^-$ and $\gamma$ rays, which can flood the detectors. Given that our experiment is carried out in an air environment, a large fraction of low-energy particles and radiation are absorbed before they can reach the screens; for instance, e$^{\pm}$ with energy $\lesssim 100$ keV are mostly absorbed by a few centimetres of air. The air environment can provide additional sources of background in the form of stray light arising from Cherenkov emission, fluorescence of air molecules, and optical transition radiation (OTR) which is produced by particles passing from different dielectric materials into the air (such as the target-air interface). While the contribution of all of these sources is small compared to the light collected due to luminescence, an aluminum blocker foil is placed in the beam path before the luminescence screen to reduce on-axis Cherenkov and OTR illuminating the Chromox screen.

## Results

The experimental results of the post-target in-beam luminescence screen are summarized in Fig. 2, comparing directly with FLUKA Monte-Carlo simulations. Figure 2a shows the raw image data of the transverse beam profile when the target is irradiated compared with when the target is removed from the proton path. The image intensity is converted to an absolute particle fluence by normalizing to the known beam density profile of the primary proton beam (Gaussian

width $\sigma_r = 1$ mm). When the target is irradiated and the secondary beam is produced, a 5-times increase in peak brightness is observed and the total intensity increases by a factor of 80. The transverse size of the beam broadens to a Lorentzian profile with half-width $\Sigma_r = 2.3$ mm due to its finite divergence (attributed to Coulomb scattering of e$^{\pm}$ with atomic nuclei in the target material).

The integrated image intensities (total intensities) obtained from 68 shots with the target irradiated (red circles) and the target removed (black diamonds) are shown in Fig. 2b, again converted to an absolute particle number. The total intensity scales with the number of protons in the primary beam. The number of e$^{\pm}$ pairs measured using the screen is $N_{\pm,\exp} = (1.02 \pm 0.05) \times 10^{13}$. This result agrees with FLUKA predictions 10 cm downstream of the target: $N_{\pm,\text{FLUKA}} = 1.04 \times 10^{13}$. The corresponding peak pair density at the screen position is $n_{\pm,\exp} = 4.5 \times 10^{11}$ cm$^{-3}$, assuming the longitudinal pair beam profile is identical to the primary beam, expected to be the case since beam elongation due to straggling in the target is calculated to be $\lesssim 5$ ps (see Supplementary Information).

An electromagnet is used to measure the energy spectra of electrons and positrons independently from the other secondaries. The total e$^{\pm}$ pair spectrum is non-thermal, characterized by a power-law distribution spanning multiple decades in energy, so it is difficult to measure the spectrum in its entirety. Furthermore, at lower energies ($E \lesssim 10$ MeV) the beam divergence of 10 s to 100 s of mrad makes it difficult to collect the whole pair population into a spectrometer and maintain a reasonable energy resolution. A practical design was chosen covering the 30–220 MeV range with an energy resolution 10–20%. This corresponds to about 40% of the pair content of the spectrum (with the fraction of the spectrum at energy <30 MeV being only 10%). This is achieved by sampling a central portion of the pair beam cross-section. Given an overall pair divergence of 15–25 mrad, the pair beam diameter at 3 m downstream is larger than the spectrometer aperture (40 mm-wide concrete aperture, 25 mm projected height on the spectrometer's screens), so the sampled beam corresponds to ~10% in particle content of the total beam. The current in the magnet coils is varied for different shots to sample different portions of the e$^-$ and e$^+$ energy spectra in the range 30–220 MeV, which are pieced together to construct the full spectrum. The electromagnet can be switched off (leaving zero remnant magnetic field) to characterize the $\gamma$-ray and hadron background. When the magnet is activated, the luminescence signal is enhanced on both

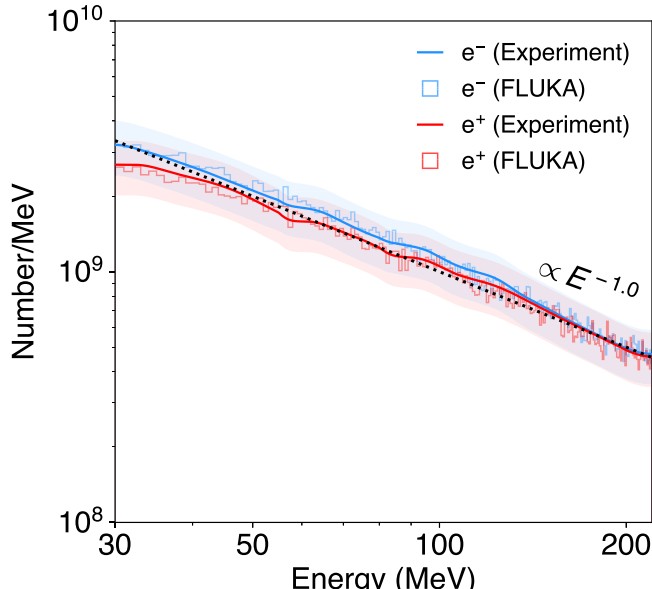

**Fig. 3 | Magnetic particle spectrometer.** The energy spectra of e⁻ (blue) and e⁺ (red) are obtained from images of luminescence screens (dimensions 200 mm × 50 mm × 1 mm) centered 240 mm from the beam axis on either side (see Fig. 1). Electrons and positrons are deflected onto the screens by the vertically-oriented dipole magnetic field of the electromagnet, whilst hadrons with a higher momentum and uncharged γ rays are mostly absorbed by the beam dump behind the electromagnet. The spectrum in the energy range $30 \leq E\,[\mathrm{MeV}] \leq 220$ is constructed by piecing together images from multiple shots using different magnetic field strengths ($B = 0.1$–$0.34$ T). The measured e⁺ spectra are characterized by a power-law with spectral index $dN/dE \propto E^{-1.0}$. The shaded regions correspond to the error associated with the absolute calibration. FLUKA simulations (histograms) are able to accurately predict the experimentally obtained spectra.

screens, which can only be explained by large numbers of e⁺ deflected onto the screens (raw data shown in the Supplementary Information). The γ rays and higher momentum hadrons in the beam are not significantly deflected and are instead absorbed by the beam dump. The main source of signal background is speckle caused by direct irradiation of the cameras with high-energy radiation scattered around the experimental tunnel. When this background is subtracted (details in the Supplementary Information), the collected segments of the spectrum overlap, and the spectra strongly resemble the FLUKA prediction (shown in Fig. 3). An absolute calibration of the e⁺ spectra is obtained by directly comparing the luminescence intensity with the screen in the direct beam path, taking the different optical setups into account (shaded regions in Fig. 3 show the error associated with this absolute calibration). The measured e⁺ spectra are characterized by a power-law with spectral index $dN/dE \propto E^{-1.0}$. This is slightly less steep than the same part of the spectrum observed at the target rear in simulations ($dN/dE \propto E^{-1.3}$) due to the sampling bias of higher energy pairs having a smaller divergence. FLUKA simulations are able to accurately predict

both the absolute number and the spectral shape of the e⁺ pairs that can reach the screens. In this energy range, the number of pairs is measured to be $N_{\pm,\mathrm{exp}} = (2.46 \pm 0.62) \times 10^{11}$ and positron fraction is $(N_{\mathrm{e}^+}/N_{\mathrm{e}^-})_{\mathrm{exp}} = 0.92 \pm 0.05$, compared with the FLUKA simulation: $N_{\pm,\mathrm{FLUKA}} = 2.45 \times 10^{11}$ and $(N_{\mathrm{e}^+}/N_{\mathrm{e}^-})_{\mathrm{FLUKA}} = 0.89$. The results of both of the luminescence screen diagnostics are summarized in Table 1.

## Discussion

Given the above agreement between the measurements and the pair beam characteristics predicted by FLUKA, the simulations used to deduce the pair spectra and peak pair density at the rear of the target are validated. Here, the full-width-half-maximum of the beam length and beam width are $\ell_{\parallel} = 17.7$ cm and $\ell_{\perp} = 0.40$ cm, respectively. For collective behavior to be observed in a plasma, the physical size of the plasma must exceed characteristic scale lengths of collective plasma processes, namely the Debye screening length, $\lambda_{\mathrm{D}}$, and the collisionless plasma skin depth, $\lambda_{\mathrm{s}}$. These quantities are traditionally defined for isotropic, thermal, non-relativistic plasmas, whereas the distribution function of e⁺ pairs produced in this work are highly anisotropic and non-thermal with relativistic thermal spreads ($k_{\mathrm{B}}T_{\pm} \gg m_{\mathrm{e}}c^2$). Calculation of the Debye screening length is thus performed in the inertial frame co-moving with the bulk of the plasma (superscript 'c.f.'), defined as the frame in which the net momentum of the beam is zero (corresponding to a bulk Lorentz factor $\Gamma_{\mathrm{bulk}} = 8$). In the co-moving frame, the Lorentz-transformed distribution function becomes approximately isotropic, although it remains non-thermal, and a Cauchy distribution provides the best fit (see the Supplementary information for details). The screening length derived for an isotropic Cauchy distribution is $\lambda_{\mathrm{D}} = \sqrt{\frac{\pi}{4}\Theta}\,\lambda_{\mathrm{s}}$, where $\lambda_{\mathrm{s}} = c/\omega_{\mathrm{p}} = \sqrt{8\pi n_{\pm}e^2/m_{\mathrm{e}}c^2}$ is the pair-plasma skin depth, and $\Theta = k_{\mathrm{B}}T_{\pm}/m_{\mathrm{e}}c^2$ (see Methods). The longitudinal and transverse thermal spreads obtained in the co-moving frame are $\Theta_{\parallel} = 6.5$ and $\Theta_{\perp} = 3.5$, so a normalized temperature $\Theta = 5.0 \pm 1.5$ is assumed to calculate the screening length in the co-moving frame.

The Lorentz transformation leads to an increase in beam length and a reduction in pair density by a factor of $\Gamma_{\mathrm{bulk}}$. Comparing the beam volume with the Debye volume,

$$\left(\frac{\ell_{\perp}^2 \ell_{\parallel}}{\lambda_{\mathrm{D}}^3}\right)^{\mathrm{c.f.}} = 5\left(\frac{N_{\pm}}{1.5 \times 10^{13}}\right)\left(\frac{n_{\pm}}{1.6 \times 10^{12}\,\mathrm{cm}^{-3}}\right)^{1/2}\left(\frac{\Gamma_{\mathrm{bulk}}}{8}\right)^{-1/2}\left(\frac{\Theta}{5}\right)^{-3/2}. \quad (1)$$

Making the same comparison with the collisionless plasma skin depth,

$$\left(\frac{\ell_{\perp}^2 \ell_{\parallel}}{\lambda_{\mathrm{s}}^3}\right)^{\mathrm{c.f.}} = 39\left(\frac{N_{\pm}}{1.5 \times 10^{13}}\right)\left(\frac{n_{\pm}}{1.6 \times 10^{12}\,\mathrm{cm}^{-3}}\right)^{1/2}\left(\frac{\Gamma_{\mathrm{bulk}}}{8}\right)^{-1/2}. \quad (2)$$

In addition, the number of particles per Debye screening volume must greatly exceed 1 for effective screening. This is indeed the case. Evaluated in the co-moving frame, $N_{\mathrm{D}}^{\mathrm{c.f.}} = (n_{\pm}\lambda_{\mathrm{D}}^3)^{\mathrm{c.f.}} = 7 \times 10^{12}$. Similarly for the collisionless skin depth, $N_{\mathrm{s}}^{\mathrm{c.f.}} = (n_{\pm}\lambda_{\mathrm{s}}^3)^{\mathrm{c.f.}} = 10^{12}$.

**Table 1 | Summary of the measured and simulated yields of electron-positron pairs**

| Position | Pair yield, $N_{\pm}$ | | Positron fraction, $(N_{\mathrm{e}^+}/N_{\mathrm{e}^-})$ | |
|---|---|---|---|---|
| | Simulation | Experiment | Simulation | Experiment |
| Target rear surface ($E \geq 1$ MeV) | $1.53 \times 10^{13}$ | - | 0.82 | - |
| Post-target screen ($E \geq 1$ MeV) | $1.04 \times 10^{13}$ | $(1.02 \pm 0.05) \times 10^{13}$ | 0.91 | - |
| Spectrometer ($30 \leq E\,[\mathrm{MeV}] \leq 220$) | $2.45 \times 10^{11}$ | $(2.46 \pm 0.62) \times 10^{11}$ | 0.89 | $0.92 \pm 0.05$ |

The experimentally measured and FLUKA-simulated electron-positron pair yield ($N_{\pm}$) and positron fraction ($N_{\mathrm{e}^+}/N_{\mathrm{e}^-}$) are summarized for the following positions: (i) at the rear surface of the target, (ii) at the post-target luminescence screen, and (iii) at the particle spectrometer luminescence screens.

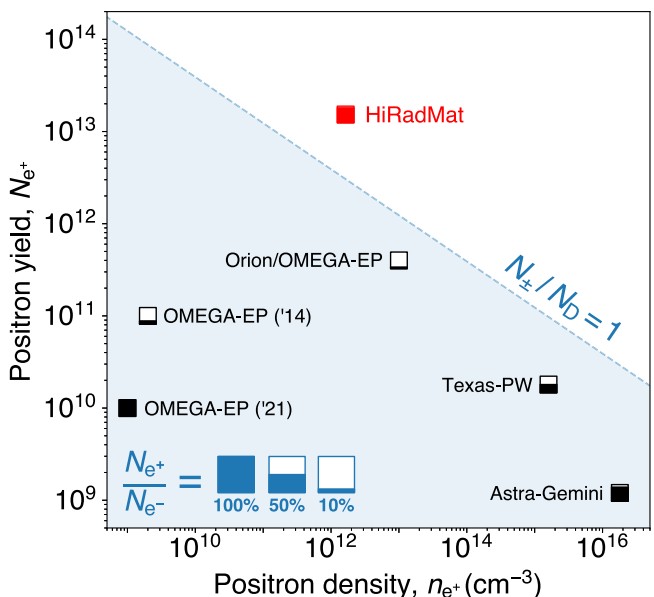

**Fig. 4 | Comparison of laboratory-produced, high-density pair beams.** The peak number and density of pairs reported in this study (red square) compared with previous experiments performed at high-power laser facilities (black squares): Orion/OMEGA-EP[15], Texas-PW[16], Astra-Gemini[17], OMEGA-EP('21)[19], OMEGA-EP('14)[30]. The data are labeled by the facility where the experiment was performed, and the fill fraction of each marker corresponds to the fraction of positrons to electrons in the experiment, $N_{e^+}/N_{e^-}$ (■ = 100%, □ = 0%, also see the key). The blue-shaded region corresponds to when the beam volume is smaller than the corresponding Debye screening volume ($N_\pm/N_D < 1$, assuming the screening length used in this work).

Furthermore, the present target configuration is not yet optimized to produce the highest possible pair yields and densities. Instead, achieving a low beam emittance was prioritized to maximize the measurable pair yield in downstream detectors. In the current setup, a large fraction of the primary protons (40%) pass through the target without significant energy loss, suggesting that using a thicker target can increase the efficiency of pair conversion (provided it can be accommodated in the experimental area). FLUKA simulations indicate that pair yields of $5 \times 10^{13}$ and densities $8 \times 10^{12}$ cm$^{-3}$ may be achievable (see Supplementary Material), making it possible to increase $(N_\pm/N_D) = (\ell_\perp^2 \ell_\parallel / \lambda_D^3)$ into the several tens, and $(N_\pm/N_s) = (\ell_\perp^2 \ell_\parallel / \lambda_s^3)$ into the several hundreds.

In Fig. 4 the number, yield, and beam neutrality of the e$^\pm$ pairs are compared with results reported in previous experiments at high-power laser facilities. It is clear that the presented scheme is able to achieve pair yields and densities in a quasi-neutral beam that will not yet be available at laser facilities without increases in laser energy by several orders of magnitude. Given that $(N_\pm/N_D) \gtrsim 1$, $(N_\pm/N_s) \gtrsim 1$ and $N_D, N_s \gg 1$ in the co-moving frame, it becomes possible for the first time to perform laboratory studies of the collective plasma behavior of relativistic electron-positron pair plasmas.

## Methods
### Electron-positron pair production target
The target is designed such that a quasi-neutral e$^\pm$ beam is produced with pair beam density maximized along a 1 m length downstream of the target. We also require that the target can be irradiated by many (potentially hundreds or thousands) of single bunches containing $O(10^{11})$ primary protons, without compromising its structural integrity. In addition, it must cool sufficiently fast to return to room temperature in-between shots, at a maximum repetition rate of 1-bunch-per-minute. The design has been optimized using two codes: (i) FLUKA[34,35] (along

with the associated interface "Flair"[36]), a particle transport Monte-Carlo scattering code capable of accurately describing the hadronic and electromagnetic cascades when the target is irradiated with 440 GeV/c protons, and (ii) Ansys® Mechanical[41], a finite-element code used to estimate the target cooling and the stress/strain induced by the energy deposition of the beam. A FLUKA-simulated transient thermal field was used as the initial condition.

FLUKA uses a robustly bench-marked physics model. To achieve a good statistical representation in the Monte-Carlo method, $10^5$ iterations were performed. The low-energy cutoff for particle transport in the simulation is 10 keV for e$^-$/e$^+$/$\gamma$ and 100 keV for hadrons.

The chosen target design consisted of a 360 mm-long cylinder of isostatic graphite (SGL Carbon R6650, 1.84 g cm$^{-3}$) and 10 mm-thick disk of tantalum, both having a 20 mm-diameter. The graphite and tantalum are housed inside a 400 mm-length, 50 mm-diameter cylinder of high-strength T9 aluminum alloy that acts as both a confinement vessel and a heat sink. The tantalum is press-fit to ensure maximal thermal contact. 2 mm-thick expanded graphite pieces (SGL Carbon Sigraflex, 1 g cm$^{-3}$) separate the target components to allow thermal expansion and reduce contact stresses during irradiation, while 2 mm-thickness Sigradur G glassy carbon beam windows are clamped onto either end of the target by aluminum flanges with Viton O-rings to hermetically seal the target materials. Using this design, the Ansys simulations have shown that the thermal loading per $3 \times 10^{11}$ protons is highest inside the tantalum, reaching peak instantaneous temperatures of 300°. Radiative and convective cooling via the outer surface of the target housing leads to cooling of the target to room temperature within a few seconds following the beam impact, while the beam-induced maximum strain of the tantalum remains, in all cases, well below its plastic deformation limit.

### Chromium-doped luminescence screens
Chromium-doped alumina-ceramic luminescence screens (Chromox, Al$_2$O$_3$: 99.5%, Cr$_2$O$_3$: 0.5%[38]) have been used to measure the particle beam intensity and transverse profile during the experiment. In the Chromox screens, principal luminescence is due to de-excitations of the lowest-excited state of Cr$^{3+}$ when energy is deposited in the screen by ionizing particles and radiation. Light is emitted isotropically, strongest at wavelengths $\lambda_1 = 691$ nm, and $\lambda_2 = 694$ nm with decay times 3–6 ms.

The transverse beam profile of the secondary beam is imaged using a 70 mm × 50 mm × 0.25 mm screen positioned 10 cm downstream of the target, and a blocker foil (50 μm aluminum) is used to minimize stray optical light. The screen is oriented at 45° to the beam path and viewed directly by a digital camera (Basler acA1920-40gm GigE camera with Sony IMX249 CMOS sensor and Canon EF 75−300 mm f/4−5.6 III lens) at a standoff distance 3.8 m with an exposure time 24 ms. An almost identical optical setup is used to image the screens in the magnetic spectrometer, except larger screens are used (200 mm × 50 mm × 1 mm, centered at a distance 240 mm off-axis), and viewed through a single mirror reflection at a standoff distance 6.2 m.

Given that relativistic particles in the energy range of interest exhibit minimum-ionizing behavior, the energy deposition of a particle passing through the Chromox screen is expected to be approximately insensitive to energy and constant between singly-charged particle species (a result confirmed by FLUKA simulations, see the Supplementary Information). The translucence of the Chromox screens to the luminescence light (attenuation length, μ = 0.8 mm$^{-1}$) limits the spatial resolution to ≥100 μm, as the luminescence light is not significantly attenuated as it is transmitted from a region where energy is deposited deeper into the screen. The translucence of the screens simplifies the analysis; as a first approximation we don't consider the different longitudinal energy deposition profiles, which are anyway shown in simulations to be approximately uniform throughout the screen thickness for the relativistic particles observed.

## Magnetic electron-positron spectrometer

Before the experiment, we characterized the spatial magnetic field profile of the electromagnet with currents supplied to the coils in the range between 0 and 400 A. The electromagnet, designated "MNPA" in the CERN internal naming system, has a yoke length of 250 mm (total length 544 mm), an aperture of width 260 mm and height 202 mm, and a maximum field at 400 A of 0.34 T. The exact magnet geometry has been modeled in detail using the finite-element code Opera 3D (Dassault Systèmes®). The field map has been calculated with sufficient resolution (10 mm) to capture the magnetic field gradients inside the magnet gap, given that the magnetic field transitions from 10–90% peak field strength over a distance 200 mm. The field map has been cross-checked against direct measurements of the magnet field using a Hall probe, and the difference between the model and measured magnetic fields inside the magnet gap is $\lesssim$2%. Finally, an energy calibration for the spectrometer (correlating screen position with $e^\pm$ energy) is obtained from particle ray-tracing calculations using the magnetic field maps. The $e^\pm$ energy ranges sampled by different magnet settings are 30–55 MeV, 55–80 MeV, 75–110 MeV, and 90–220 MeV.

An absolute calibration of the electron and positron numbers is made by using the brightness (pixel counts per unit area) of the in-beam luminescence screen. Specifically, we account for the difference between the amount of light collected in the optical setups used for the in-beam screen luminescence and for the spectrometer screens, considering (i) the different standoff distances, which leads to different solid angle subtended ($d\Omega_{\text{beam}}/d\Omega_{\text{spect}} = 2.7 \pm 0.4$); (ii) the different camera gain settings used ($G_{\text{spect}} = 38 \pm 2$, $G_{\text{beam}} = 1$); and (iii) the different thickness of screen used, where thicker screens le`ad to larger energy deposition per particle ($u_{\text{dep,spect}}/u_{\text{dep,beam}} = 3.5 \pm 0.5$). A small $1/\cos\theta$ geometric correction is applied to the energy spectra to account for the additional path length of Chromox encountered by obliquely-incident deflected particles, where $\theta = 10°$ at the screen edge closest to the beam axis, and $\theta = 25°$ at the furthest edge.

The digital cameras viewing the in-beam screen at a standoff distance of 3.8 m can resolve features as small as 50 μm in size, whilst the cameras viewing the spectrometer screens at a standoff distance 6.2 m can resolve features 120 μm in size. However, the resolution of the energy spectrum projected onto the spectrometer screens is limited by the 20cm-thickness, 40 mm-wide concrete aperture at the entrance of the electromagnet. Since high-Z collimators can cause unwanted shaping of the spectrum by inducing further scattering and conversion, concrete is chosen as the shielding material.

In our experimental setup, the target has been placed on a vertically movable, high-precision stage, allowing us to acquire data with the target in-beam, as well as in a 'target-out' position. In the latter case, the primary proton beam continues at its full intensity through the luminescence screens and the electromagnet towards the beam dump. Using the 'target-out' configuration, we took measurements without the current supplied to the electromagnet to characterize the hadron and lepton background produced as particles are back-scattered by the proton beam impact on the beam dump. The remnant field of the aforementioned electromagnet was measured extensively before the experiment using a Hall probe and was found reproducibly to be negligible (on the order of the noise of the instrument, i.e., $B \lesssim 0.3$ mT).

## Screening lengths of relativistic plasmas

The plasma screening length, $\lambda_D$, is obtained by evaluating the static ($\omega \to 0$), long wavelength ($k \to 0$) limit of the dielectric tensor[42–44]:

$$\lambda_D^{-2} = \lim_{k\to 0} \lim_{\omega/k \to 0} k^2(\varepsilon_\ell - 1), \qquad (3)$$

where $\varepsilon_\ell$ is the longitudinal component of the dielectric tensor. Written in terms of the dimensionless momentum $\mathbf{u} = \mathbf{p}/m_\alpha c = \gamma\boldsymbol{\beta}$,

$$\varepsilon_\ell = 1 + \sum_\alpha \frac{\omega_{p\alpha}^2}{c^2 k^2} \int_{C_L} \frac{1}{\omega - c\mathbf{k}\cdot\boldsymbol{\beta}} c\mathbf{k}\cdot\frac{\partial f_\alpha(\mathbf{u})}{\partial \mathbf{u}} d^3u, \qquad (4)$$

integrated along the Landau contour, $C_L$, and summing the contribution from each species component of the plasma, $\alpha$. The plasma frequency associated with each species is $\omega_{p\alpha} = (4\pi n_\alpha q_\alpha^2/m_\alpha c^2)^{1/2}$, and $f_\alpha$ is the distribution function. If the distribution function is isotropic, the screening length is simply evaluated:

$$\lambda_D^{-2} = -4\pi \sum_\alpha \frac{\omega_{p\alpha}^2}{c^2} \int_0^\infty \gamma u \frac{\partial f_\alpha(u)}{\partial u} du. \qquad (5)$$

The well-known Debye screening length for a non-relativistic plasma is obtained when a Maxwellian distribution is assumed: $\lambda_D = \Theta\lambda_s$, where $\Theta = k_B T/m_\alpha c^2$ is the temperature normalized to the electron mass, and $\Theta \ll 1$. Assuming instead a plasma with a relativistic temperature ($\Theta \gtrsim 1$), a commonly used distribution function is the relativistic Maxwellian (Jüttner-Synge) distribution. In this case ($\Theta \gtrsim 1$):

$$f_\alpha(u) = \frac{\exp(-\gamma/\Theta)}{4\pi\Theta K_2(1/\Theta)} \quad \Rightarrow \quad \lambda_D = \sqrt{\Theta}\,\lambda_s, \qquad (6)$$

where $K_2(1/\Theta)$ is a modified Bessel function of the second kind of order 2.

Since the electron-positron distribution functions are non-thermal with high-energy tails characterized by power-law distributions ($f_\alpha(u) \propto \gamma^{-m}$, $m \approx 1$–2), a better fit is obtained by assuming a relativistic Cauchy distribution:

$$f_\alpha(u) = \frac{\Theta}{\pi^2(\Theta^2 + u^2)^2} \quad \Rightarrow \quad \lambda_D = \sqrt{\frac{\pi}{4}}\Theta\lambda_s. \qquad (7)$$

The two screening lengths are similar because the distribution functions only differ in the very high-energy range. The relativistic Cauchy distribution is fitted to the particle momentum distributions in the inertial frame co-moving with the bulk to obtain the transverse temperature $\Theta_\perp = 3.5$ and the longitudinal temperature $\Theta_\parallel = 6.5$ (details of the fitting are given in the Supplementary Information). A temperature $\Theta = 5.0 \pm 1.5$ is assumed to calculate the plasma screening length derived for an isotropic Cauchy distribution.

## Data availability
The data used in this study are available in the public repository: https://doi.org/10.5281/zenodo.11190804.

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

## Acknowledgements

We thank Professor C. Joshi (UCLA), Dr. F. Albert (LLNL), and Dr. C. Densham (STFC Rutherford Appleton Laboratory) for useful discussions, as well as Mr. T. E. Levens (CERN) and Dr. T. Ma (LLNL) for supporting this experiment. This project has received funding from the European Union's Horizon Europe Research and Innovation program under Grant Agreement No 101057511 (EURO-LABS). The work of G.G. was partially supported by UKRI under grant no. ST/W000903/1 and EP/Y035038/1, while A.F.A.B. was also supported by UKRI (grant number MR/W006723/1). The work of D.H.F. and D.H. was supported by the U.S. Department of Energy under Award Number DE-NA0004144. We also acknowledge funding from AWE plc., and the Central Laser Facility (STFC). FLUKA simulations were performed using the STFC Scientific Computing Department's SCARF cluster. UK Ministry of Defence ⓒ Crown Owned Copyright 2024/AWE.

## Author contributions

This project was conceived by G.G. and R.B. The experiment was designed by C.D.A., G.G., P.S., N.C., and R.B. and carried out by C.D.A., P.S., N.C., G.G., T.H., R.S., J.W.D.H., P.J.B., S.B., F.D.C., A.G., D.H., S.I., V.S., T.V., and B.T.H. The data analysis was carried out by C.D.A. The manuscript was written by C.D.A., with input from G.G., R.B., N.C., L.O.S., and B.R. Numerical simulations were performed by C.D.A. and P.S. Further experimental and theoretical support was provided by I.E., D.H.F., A.F.A.B., A.A.S., J.T.G., B.T.H., F.M., S.S., B.R., H.C., L.O.S., T.D. and R.M.G.M.T.

## Competing interests

The authors declare no competing interests.

## Additional information

[1]Department of Physics, University of Oxford, Parks Road, Oxford OX1 3PU, UK. [2]European Organization for Nuclear Research (CERN), CH-1211 Geneva 23, Switzerland. [3]GSI Helmholtzzentrum für Schwerionenforschung GmbH, Planckstraße 1, 64291 Darmstadt, Germany. [4]GoLP/Instituto de Plasmas e Fusão Nuclear, Instituto Superior Técnico, Universidade de Lisboa, 1049-001 Lisboa, Portugal. [5]Lawrence Livermore National Laboratory, 7000 East Ave, Livermore, CA 94550, USA. [6]STFC Rutherford Appleton Laboratory, Chilton, Didcot OX11 0QX, UK. [7]University of Rochester Laboratory for Laser Energetics, Rochester, NY 14623, USA. [8]Science Institute, University of Iceland, Dunhaga 3, IS-107 Reykjavik, Iceland. [9]Division of Space and Plasma Physics, School of Electrical Engineering and Computer Science, KTH Royal Institute of Technology, SE-100 44 Stockholm, Sweden. [10]AWE, Aldermaston, Reading, Berkshire RG7 4PR, UK. [11]Max-Planck-Institut für Kernphysik, Saupfercheckweg 1, D-69117 Heidelberg, Germany. [12]School of Applied Mathematics and Physical Sciences, National Technical University of Athens, Athens 157 72, Greece. [13]Department of Physics, University of Strathclyde, Glasgow G4 0NG, UK. ✉e-mail: charles.arrowsmith@physics.ox.ac.uk

