## [Peer Review File · Nature Communications]

Laboratory realization of relativistic pair-plasma beamsReviewers' comments:

Reviewer #1 (Remarks to the Author):

Dear Editor,

I have carefully read the manuscript "Laboratory realization of relativistic pair-plasma beams" submitted by Arrowsmith et al. for publication in Nature Communications (Manuscript NCOMMS-23-63777-T). My overall impression of this article is extremely favorable, and I would like to recommend its acceptance after a minor revision in response to my comments below. The methodology of this study is sound, easily meeting the standards accepted in the field, and the results are significant and original.

This pioneering article reports an important breakthrough in laboratory plasma physics --- the production, perhaps for the first time, of a macroscopic relativistic electron-positron plasma (with size greater than the plasma Debye length) in the lab. Such plasmas are ubiquitous in high-energy astrophysics, found, for example, around neutron stars (e.g., in pulsar and magnetar magnetospheres, pulsar winds, and pulsar wind nebulae such as the Crab Nebula) and probably in relativistic jets from rotating black holes in X-ray binaries and active galactic nuclei, as well as, perhaps, in gamma-ray burst jets. Consequently, their behavior is of great interest in astrophysics and has been studied extensively using analytical theory and numerical simulations. At the same time, however, experimental laboratory studies of pair-plasma dynamics have not been possible until now, for the simple reason that such plasmas have not yet been created in the lab. This is in stark contrast to numerous experimental studies of plasma processes in standard, nonrelativistic electron-ion plasmas, conducted over the last few decades, which have been very instrumental and influential and have led, in combination with theory and simulation, to real progress in our understanding of fundamental plasma processes, with important applications to space and solar environments. A critical aspect of the remarkable experimental achievement reported in this article is the creation of not just a large number of electrons and positrons, but of a dense and compact enough cloud of pairs to ensure that its size significantly exceeds the characteristic plasma microscales such as the Debye length and the collisionless skin depth. This is necessary in order for such a plasma to be able to exhibit collective plasma behavior --- a hallmark of plasma physics. It is this aspect of the study that makes it stand out among other recent pair-production experiments, as it demonstrates the creation of a novel experimental platform that promises to make laboratory studies of collective plasma phenomena in relativistic pair plasmas possible for the first time. This, in turn, opens for exploration a new, exciting frontier in laboratory plasma astrophysics.

I would like to ask the Authors to address the following comments, almost of them relating to page 5 of the main part of the manuscript:

1) Last paragraph on page 4, first sentence, lines 136-137: "An electromagnet is used to measure the particle energy spectra ...". I find this statement a little bit misleading. If I understand correctly, while it is not obvious at this point in the narrative, what the electromagnet measures is not the entire distribution function but actually just the high-energy (> 30 MeV) tail of the produced pair distribution, well above the peak. This can cause some confusion. I suggest clarifying this explicitly, making this sentence more precise, e.g., by inserting the words "the high-energy part of the ..." after "is used to measure". This can also be explained later in the paper, somewhere on page 5. I think such a clarification is important because it highlights the critical role of numerical simulations in interpreting the experimental results. If I understand correctly, since we cannot access/measure directly the bulk/core of the produced pair distribution function, we use the simulations (where we can measure everything and everywhere), benchmarked by comparison with the experimentally measurable high-energy part of the distribution, to deduce anything we want to know about the produced pair population, including its overall density. I think this point is important for the reader to understand

and thus needs to be made clearer.

2) Page 5, Fig 3: the plotted spectra in this log-log plot seem to be well approximated by a straight line, indicating a power-law (as opposed to thermal) spectrum. This is in clear contradiction with the statements in the paper, e.g., in the 3rd paragraph on page 5, that the particle population can be modeled by a drifting (Lorentz-boosted) relativistic Maxwellian distribution. This striking contradiction is somewhat reconciled later on, in the very last section of the Supplementary Material (SM), where it becomes clear (e.g., Fig. 4 on page 3 of SM) that there is a thermal bulk of the Lorentz-transformed particle distribution, which is well fit by a relativistic Maxwellian (with thermal $\Gamma_T = 8$) that, however, does not quite describe the high-energy power-law tail plotted in Fig. 3 on page 5. The problem, however, is that when the Reader is reading page 5 on the Main text, he or she does not yet know of the existence of that section of the SM. This causes major confusion and leads to undue skepticism. I would recommend addressing this issue by adding a forward reference to that last section of the SM and a sentence (somewhere on page 5 or in the Caption for Fig. 3) clarifying the difference between the obvious high-energy power law seen in Fig. 3 and the main, lower-energy ($T=4$ MeV) thermal bulk discussed in the text.

3) In any case, since there is such a clear nonthermal power-law distribution (straight line) in Fig. 3, the reader will naturally wonder what the corresponding power-law index is. Please quote a value for this index, even if approximate. And I would also suggest adding a straight dashed line in Fig. 3 representing the best power-law fit.

4) Page 5, second paragraph, last sentence (lines 163-165): I suspect that there is some terminology confusion here, which can be easily corrected. The condition for the plasma to be able to exhibit collective phenomena is indeed that the size L of the plasma is greater than the Debye length, and this condition can indeed be recast in the form $N_{\text{pm}} / N_D \gg 1$, as displayed at the end of this sentence (line 165), since $N_{\text{pm}} = n_{\text{pm}} L^3$ and $N_D = n_{\text{pm}} r_{\text{scr}}^3$. But this condition is not at all equivalent to the condition that the number of particles per Debye screening volume is much greater than one! That condition is just $N_D \gg 1$, which is independent of system size L . These two conditions are completely independent of each other and should be confused. I suggest correcting the wording carefully, e.g., by just removing the phrase “the number of particles per Debye screening volume is much greater than one” at the end of this paragraph (lines 164-165). A similar change should then also be made at the end of the next paragraph, just above Equation (1) (line 175): the ratio N_{pm}/N_D is indeed the quantity of interest here, but it is not the number of particles in a Debye screening volume. A corresponding correction should also be made in the last section of the SM. If the Authors do want to mention the condition $N_D \gg 1$ somewhere, they should discuss it separately from the conditions $L \gg r_{\text{scr}}$ or $N_{\text{pm}} \gg N_D$.

5) A small cosmetic question; why is the Debye screening length denoted by r_{scr} . This notation seems rather unconventional to me. Usually, this quantity is denoted by λ_D .

6) Page 5, 3rd paragraph (“The pair beams produced ...”): there is a sentence here about “approximately isotropic momentum distribution” in the boosted frame, fitted with a relativistic Maxwellian with $T=4$ MeV (lines 171-173). Once again, why should the boosted particle population be expected to be approximately isotropic and thermal? This is not at all obvious (especially in light of the markedly nonthermal spectrum shown in Fig. 3). This question is, in fact, addressed in the last section of SM, but, once again, the reader at this point in the narrative does not yet know that such a section even exists, does not have a way of anticipating it. It would be good to add here a forward reference to that section, pointing the puzzled reader there for details. And perhaps once again say that that thermal spectrum describes the low/average energy range and does, not for example, describe the high-energy spectrum shown in Fig. 3. In addition, it would be useful, if allowed, to number the different sections of the SM, this would make it easier to refer to them.

7) The third paragraph on page 5 claims that the comoving distribution is isotropic, and this is indeed

illustrated in panels b and c in Fig. 3 in the SM, which show that the longitudinal and transverse temperatures are equal. I think this is a nontrivial result, it is not a priori clear why this should be the case. I therefore think it is worth highlighting. Likewise, the result the central core of the distribution function is well approximated by a Maxwellian (although it looks like there are also nonthermal tail) is also nontrivial and should be stressed.

8) The third paragraph on page 5, culminating with Equation (1), ends rather abruptly, without stating the main conclusion. The reader naturally want to ask, "so what?" I would suggest editing the end of this paragraph add a concluding statement briefly summarizing the main finding of this paragraph, e.g., the statement that the size of the created pair plasma is greater than the Debye length and so such a plasma is capable of exhibiting collective behaviour.

9) The last paragraph on page 5 states that "the beam size must exceed ... the plasma skin depth of the ambient plasma ...", and then goes on to reformulating this condition in terms of the ratio λ_{pm}/N_s . But wouldn't be of interest to compare the relevant length scales directly? I think the reader would benefit from knowing what that plasma skin depth is and what the longitudinal and transverse beam sizes are (in either frame). Please give this information. In addition, my Comment 8 above applies to this paragraph as well: it does not have a good ending, just ends with some boring estimate without formulating a clear conclusion or insight derived from it. This just provokes the question "And so what?"

10) This last point applies again to the first paragraph on page 6 ("In Figure 4 we plot ..."). There is no clear narrative here, it is not clear what point the Authors are trying to make. The argument is just dropped midway, without reaching a conclusion. Are you arguing that your CERN experiment is more advantageous, in some sense, than the laser-based experiments? If so, this claim needs to be clearly and explicitly stated.

11) End of page 6: The last sentence of this main part of the paper presents the main claim of this paper --- that these experiments have successfully produced a macroscopic relativistic pair plasma which can now be used as a laboratory platform for studies of collective plasma processes in this astrophysically important regime. This is a good sentence, and I am not disputing this claim, but to me it seemed somewhat dry and not sufficiently strong. I would suggest making the ending of the paper more exciting and uplifting, perhaps expanding on the profound implications of this research, its astrophysical significance. E.g., one could stress that this study opens up a new, exciting direction/frontier in relativistic plasma astrophysics, and perhaps speculate a little bit on how exactly it paves the way for future studies, what kinds of important collective plasma processes can be studied with this approach in the near future.

I hope the Authors will find these comments useful and that addressing them will help improve this already very significant paper.

Dmitri Uzdensky

Reviewer #2 (Remarks to the Author):

The manuscript reports on the generation of an electron-positron pair beam using the 440 GeV/c proton beam from Super Proton Synchrotron at CERN, by using hadronic and electromagnetic cascades in graphite and tantalum targets, respectively. The total number of pairs and the fluence of the pair beam are measured experimentally at the location of the post-target profile monitor (10 cm downstream of the target exit), and the pair beam density is deduced assuming it has the same duration as the proton beam. The energy spectrum of the pairs is also measured, in the spectral range from 30 to 220 MeV, corresponding to about 2.5% of the pairs in the beam and thus representing mostly the energy tail of the pair beam and not the bulk of the pair beam. These experimental

measurements are in very good agreement with FLUKA simulations modelling the hadronic and electromagnetic cascades. FLUKA simulations are then used to extrapolate the experimental measurements at the location of the target exit, and to provide critical physical quantities to assess the plasma nature of the pair beam, such as the Lorentz factor of the transformation from the laboratory to the zero-momentum frame ($\Gamma_0 = 4.5$) and the pair beam temperature (4 MeV). Using these numerical quantities, it is then concluded that the pair beam can be considered a plasma as its size is larger than the Debye length.

This work certainly represents an important step in an experimental project aiming at the study of collective plasma behaviours with pair beams, but its scientific impact is limited as the plasma nature of the pair beam is not demonstrated experimentally. No plasma collective effect is reported in the manuscript, which would represent the most convincing demonstration that the pair beam is indeed a plasma. The experimental measurement of pair density is not used, instead it is the extrapolated value from the FLUKA simulation, and Γ_0 and the pair temperature are entirely coming from the simulations. For example, in the Supplementary information the authors argue that the bulk of the pair beam in the longitudinal momentum distribution is the most important when assessing the collective behaviour of a plasma (Fig. 4 in SI shows that the most important particles are in the range 0-40 MeV), and not the extended spectral tail. Yet the experimental measurement misses the bulk and only characterise 2.5% of the pairs in the spectral tail (30-220 MeV). The main conclusion of the manuscript, namely the plasma nature of the generated pair beam, is thus not substantiated by the experimental measurements but mostly by the FLUKA simulations.

The work deserves to be published with the improvements suggested below, but given its limited scientific impact it is more appropriate for a more specialised journal, and I therefore do not recommend its publication in Nature Communications.

List of comments:

- 1) Lines 37-38 in the abstract. The sentence "The produced pair beams have a volume that fills multiple Debye spheres and are thus able to sustain collective plasma oscillations" needs to be reformulated to clearly state what data supports this conclusion, and what comes from experiment and what comes from simulation. This is critical for the soundness of the paper: it must be stated clearly how the conclusion is substantiated by the data.
- 2) Lines 185-186 in the conclusion. "the electron-positron beam we produced at HiRadMat can safely be assumed to behave as a relativistic pair plasma": same comment as point 1), clearly state how this conclusion is substantiated by the data, and the type of data (experiment versus simulation) used to support this conclusion.
- 3) Line 144. "FLUKA simulations predict that without the magnetic field, low energy ($\lesssim 10$ MeV) scattered e^\pm can irradiate the screens, but when the electromagnet is activated, these lower-momentum pairs are swept away and the measured signals more closely resemble the true spectra." Without magnetic field, the device is not measuring spectra at all (we can't say that magnetic field improves and make the measurement resembles more closely the true spectrum, it is simply not a spectrum without B field). This confusing sentence needs to be reformulated, and to state explicitly how the spectrum is obtained with regards to a possible background subtraction. Do the authors subtract the magnet-off case from the magnet-on data, or oppositely do not subtract the magnet-off data because it is assumed that this background disappears when the magnet is turned on?
- 4) Lines 161-162. "We assess whether this density is sufficiently high for the pair beam to be considered a plasma": the pair density is not the only important parameter for getting a plasma, as explained in the next sentence and as shown in Fig. 4. Something like "We assess whether such a pair beam can be considered a plasma" is more appropriate.

5) Lines 164-165. "if the number of particles per Debye screening volume is much greater than one": the number of particles per Debye screening volume is N_D , and it's the total number of pairs N_{pairs} that needs to be much greater than N_D . This sentence should be corrected.

Reviewer #3 (Remarks to the Author):

The authors have presented very exciting work, where, for the first time, enough electrons and positrons have been created simultaneously to open up the study of collective physics in electron-positron pair plasmas. Simulation results agree well with the experimental findings, assuring that the conclusions of the manuscript are well-founded. I think that the paper constitutes a breakthrough, and recommend publication enthusiastically.

3rd April 2024

Response to Reviewers for manuscript NCOMMS-23-63777A-Z

We sincerely thank all three reviewers for their feedback. Their comments have helped in significantly improving the manuscript. Our reply to each of their queries is detailed below. The corresponding changes to the manuscript have been highlighted using a red font in the revised manuscript.

Reviewer #1 (Remarks to the Author):

Dear Editor,

I have carefully read the manuscript “Laboratory realization of relativistic pair-plasma beams” submitted by Arrowsmith et al. for publication in Nature Communications (Manuscript NCOMMS-23-63777-T). My overall impression of this article is extremely favorable, and I would like to recommend its acceptance after a minor revision in response to my comments below. The methodology of this study is sound, easily meeting the standards accepted in the field, and the results are significant and original.

This pioneering article reports an important breakthrough in laboratory plasma physics --- the production, perhaps for the first time, of a macroscopic relativistic electron-positron plasma (with size greater than the plasma Debye length) in the lab. Such plasmas are ubiquitous in high-energy astrophysics, found, for example, around neutron stars (e.g., in pulsar and magnetar magnetospheres, pulsar winds, and pulsar wind nebulae such as the Crab Nebula) and probably in relativistic jets from rotating black holes in X-ray binaries and active galactic nuclei, as well as, perhaps, in gamma-ray burst jets. Consequently, their behavior is of great interest in astrophysics and has been studied extensively using analytical theory and numerical simulations. At the same time, however, experimental laboratory studies of pair-plasma dynamics have not been possible until now, for the simple reason that such plasmas have not yet been created in the lab. This is in stark contrast to numerous experimental studies of plasma processes in standard, nonrelativistic electron-ion plasmas, conducted over the last few decades, which have been very instrumental and influential and have led, in combination with theory and simulation, to real progress in our understanding of fundamental plasma processes, with important applications to space and solar environments. A critical aspect of the remarkable experimental achievement reported in this article is the creation of not just a large number of electrons and positrons, but of a dense and compact enough cloud of pairs to ensure that its size significantly exceeds the characteristic plasma microscales such as the Debye length and the collisionless skin depth. This is necessary in order for such a plasma to be able to exhibit collective plasma behavior --- a hallmark of plasma physics. It is this aspect of the study that makes it stand out among other recent pair-production experiments, as it demonstrates the creation of a novel experimental platform that promises to make laboratory studies of collective plasma phenomena in relativistic pair plasmas possible for the first time. This, in turn, opens for exploration a new, exciting frontier in laboratory plasma astrophysics.

I would like to ask the Authors to address the following comments, almost of them relating to page 5 of the main part of the manuscript:

1) Last paragraph on page 4, first sentence, lines 136-137: “An electromagnet is used to measure the particle energy spectra ...”. I find this statement a little bit misleading. If I understand correctly, while it is not obvious at this point in the narrative, what the electromagnet measures is not the entire distribution function but actually just the high-energy (> 30 MeV) tail of the produced pair distribution, well above the

peak. This can cause some confusion. I suggest clarifying this explicitly, making this sentence more precise, e.g., by inserting the words “the high-energy part of the ...” after “is used to measure”. This can also be explained later in the paper, somewhere on page 5. I think such a clarification is important because it highlights the critical role of numerical simulations in interpreting the experimental results. If I understand correctly, since we cannot access/measure directly the bulk/core of the produced pair distribution function, we use the simulations (where we can measure everything and everywhere), benchmarked by comparison with the experimentally measurable high-energy part of the distribution, to deduce anything we want to know about the produced pair population, including its overall density. I think this point is important for the reader to understand and thus needs to be made clearer.

- We thank the reviewer for pointing out the confusion about the portion of the pair spectrum that is measured using the electron-positron spectrometer. In fact, the measurement of the electron-positron pairs in the energy range 30-220 MeV corresponds to approximately 40% of the total pair spectrum, therefore corresponding to a significant proportion of the bulk rather than just the high energy tail. We agree that this was not made clear enough in the original manuscript and we have added clarification in the revised manuscript. In addition, we added an explanation that the multi-decadal nature of the non-thermal spectrum makes it practically impossible to measure in its entirety, and why the measurement of pairs at lower energies (<30 MeV) becomes progressively difficult due to their larger divergence.
- Indeed it is an important point to make clear that the reason for using the FLUKA simulations is because we cannot directly measure certain characteristics of the produced pair population. We have made this point clearer.

2) Page 5, Fig 3: the plotted spectra in this log-log plot seem to be well approximated by a straight line, indicating a power-law (as opposed to thermal) spectrum. This is in clear contradiction with the statements in the paper, e.g., in the 3rd paragraph on page 5, that the particle population can be modeled by a drifting (Lorentz-boosted) relativistic Maxwellian distribution. This striking contradiction is somewhat reconciled later on, in the very last section of the Supplementary Material (SM), where it becomes clear (e.g., Fig. 4 on page 3 of SM) that there is a thermal bulk of the Lorentz-transformed particle distribution, which is well fit by a relativistic Maxwellian (with thermal $\Gamma_T = 8$) that, however, does not quite describe the high-energy power-law tail plotted in Fig. 3 on page 5. The problem, however, is that when the Reader is reading page 5 on the Main text, he or she does not yet know of the existence of that section of the SM. This causes major confusion and leads to undue skepticism. I would recommend addressing this issue by adding a forward reference to that last section of the SM and a sentence (somewhere on page 5 or in the Caption for Fig. 3) clarifying the difference between the obvious high-energy power law seen in Fig. 3 and the main, lower-energy ($T=4$ MeV) thermal bulk discussed in the text.

- We agree with the reviewer that fitting the pair population to a thermal distribution is a cause of confusion because the measured spectrum is so clearly a non-thermal power-law distribution. As pointed out by the reviewer, the relativistic Maxwellian is only a good fit to the pair distribution in the thermal bulk, therefore we have now considered a Cauchy distribution (described in Methods). The Cauchy distribution provides a better fit to the high-energy tail as well as the thermal bulk (corresponding to >98.5% of the pair spectrum). A comparison of the Cauchy and relativistic Maxwellian fits is shown in Figure 7 of the Supplementary Material, where it is clear that the Cauchy distribution is a far more convincing fit. Interestingly, when the Debye screening length is re-derived assuming the Cauchy distribution, we find that it gives only a slight modification from the screening length of a relativistic Maxwellian (by only ~10%, as discussed in the Supplementary Information). Though this is not a totally unexpected result since the two distribution functions are approximately identical in their more populated thermal bulk.
- Forward references to the Methods and Supplementary Information have been added to the main text to make it clearer that additional technical details can be found in these sections.

3) In any case, since there is such a clear nonthermal power-law distribution (straight line) in Fig. 3, the reader will naturally wonder what the corresponding power-law index is. Please quote a value for this index, even if approximate. And I would also suggest adding a straight dashed line in Fig. 3 representing the best power-law fit.

- We agree with this comment and we have added a dashed line to Figure 3 showing the power-law index of the spectrum. A comment has been added to the main text to point out that the measured spectrum is characterized by a power-law distribution, with a clarification that sampling bias of the spectral measurement leads to a slightly shallower spectrum than is found at the immediate rear of the target. In a revised Figure 1 of the Supplementary Information, power-law fits are also included for the FLUKA simulation of the electron-positron spectrum at the target rear.

4) Page 5, second paragraph, last sentence (lines 163-165): I suspect that there is some terminology confusion here, which can be easily corrected. The condition for the plasma to be able to exhibit collective phenomena is indeed that the size L of the plasma is greater than the Debye length, and this condition can indeed be recast in the form $N_{\text{pm}} / N_D \gg 1$, as displayed at the end of this sentence (line 165), since $N_{\text{pm}} = n_{\text{pm}} L^3$ and $N_D = n_{\text{pm}} r_{\text{scr}}^3$. But this condition is not at all equivalent to the condition that the number of particles per Debye screening volume is much greater than one! That condition is just $N_D \gg 1$, which is independent of system size L . These two conditions are completely independent of each other and should be confused. I suggest correcting the wording carefully, e.g., by just removing the phrase “the number of particles per Debye screening volume is much greater than one” at the end of this paragraph (lines 164-165). A similar change should then also be made at the end of the next paragraph, just above Equation (1) (line 175): the ratio N_{pm}/N_D is indeed the quantity of interest here, but it is not the number of particles in a Debye screening volume. A corresponding correction should also be made in the last section of the SM. If the Authors do want to mention the condition $N_D \gg 1$ somewhere, they should discuss it separately from the conditions $L \gg r_{\text{scr}}$ or $N_{\text{pm}} \gg N_D$.

- We thank the reviewer for pointing this out. We have removed the phrasing “the number of particles per Debye screening volume is much greater than one”, referring instead to the more precise statement $N_{\text{pm}} / N_D > 1$, which is defined in terms of the beam dimensions and the screening length.
- The condition $N_D \gg 1$ is included as a separate discussion.

5) A small cosmetic question; why is the Debye screening length denoted by r_{scr} . This notation seems rather unconventional to me. Usually, this quantity is denoted by λ_D .

- We agree with the reviewer that the notation λ_D is more familiar to the community so we have made the change to the text throughout.

6) Page 5, 3rd paragraph (“The pair beams produced ...”): there is a sentence here about “approximately isotropic momentum distribution” in the boosted frame, fitted with a relativistic Maxwellian with $T=4$ MeV (lines 171-173). Once again, why should the boosted particle population be expected to be approximately isotropic and thermal? This is not at all obvious (especially in light of the markedly nonthermal spectrum shown in Fig. 3). This question is, in fact, addressed in the last section of SM, but, once again, the reader at this point in the narrative does not yet know that such a section even exists, does not have a way of anticipating it. It would be good to add here a forward reference to that section, pointing the puzzled reader there for details. And perhaps once again say that that thermal spectrum describes the low/average energy range and does, not for example, describe the high-energy spectrum shown in Fig. 3. In addition, it would be useful, if allowed, to number the different sections of the SM, this would make it easier to refer to them.

- Following from the response of Comment #2 where the improved fitting to a Cauchy distribution was discussed, it should now be clearer in the text and the Supplementary Information that the relativistic Maxwellian is a reasonable fit only to the low-energy thermal bulk. Clarifications have been added to the main text to re-state that the distribution is non-thermal.
- Furthermore, from the improved fitting it should now be clearer that in the co-moving frame the pair distribution is almost (but not quite) isotropic, and fitted temperatures have been provided for the transverse and longitudinal distributions separately (see Supplementary Information). It is made clear that a single value for the normalized temperature is assumed based on these fitted temperatures which allows the screening length derived for an isotropic Cauchy distribution.
- To make it easier for the reader to locate the relevant sections of the Supplementary Information, a contents section (with hyperlinks) has been added.

7) *The third paragraph on page 5 claims that the comoving distribution is isotropic, and this is indeed illustrated in panels b and c in Fig. 3 in the SM, which show that the longitudinal and transverse temperatures are equal. I think this is a nontrivial result, it is not a priori clear why this should be the case. I therefore think it is worth highlighting. Likewise, the result the central core of the distribution function is well approximated by a Maxwellian (although it looks like there are also nonthermal tail) is also nontrivial and should be stressed.*

- As discussed in the response to the previous comment, it should now be clearer in the main text and the Supplementary Information that the co-moving distribution is approximately isotropic and separate fitted temperatures for the longitudinal and transverse distributions are now provided.

8) *The third paragraph on page 5, culminating with Equation (1), ends rather abruptly, without stating the main conclusion. The reader naturally want to ask, “so what?” I would suggest editing the end of this paragraph add a concluding statement briefly summarizing the main finding of this paragraph, e.g., the statement that the size of the created pair plasma is greater than the Debye length and so such a plasma is capable of exhibiting collective behaviour.*

- We agree with this comment that the comparison of the size of the created pair plasma with the Debye length requires a concluding remark, and so we have added the statement that given $N_{\perp}/N_D \gtrsim 1$, $N_{\parallel}/N_s \gtrsim 1$ and $N_D \gg 1$, it becomes possible for the first time to perform laboratory studies of the collective plasma behaviour of relativistic electron-positron pair plasmas.

9) *The last paragraph on page 5 states that “the beam size must exceed ... the plasma skin depth of the ambient plasma ...”, and then goes on to reformulating this condition in terms of the ratio N_{\perp}/N_s . But wouldn't be of interest to compare the relevant length scales directly? I think the reader would benefit from knowing what that plasma skin depth is and what the longitudinal and transverse beam sizes are (in either frame). Please give this information. In addition, my Comment 8 above applies to this paragraph as well: it does not have a good ending, just ends with some boring estimate without formulating a clear conclusion or insight derived from it. This just provokes the question “And so what?”*

- We agree that it would be of interest to compare the relevant length scales directly and we have now included this information.

10) *This last point applies again to the first paragraph on page 6 (“In Figure 4 we plot ...”). There is no clear narrative here, it is not clear what point the Authors are trying to make. The argument is just dropped midway, without reaching a conclusion. Are you arguing that your CERN experiment is more advantageous, in some sense, than the laser-based experiments? If so, this claim needs to be clearly and explicitly stated.*

- We agree with this comment that the discussion of Figure 4 is lacking a conclusion statement to make its relevance clear. To address this, we have added the comment that the presented scheme is able to achieve pair yields and densities in a quasi-neutral beam that will not be achievable at laser facilities without several orders of magnitude increase in laser energy.

11) *End of page 6: The last sentence of this main part of the paper presents the main claim of this paper --- that these experiments have successfully produced a macroscopic relativistic pair plasma which can now be used as a laboratory platform for studies of collective plasma processes in this astrophysically important regime. This is a good sentence, and I am not disputing this claim, but to me it seemed somewhat dry and not sufficiently strong. I would suggest making the ending of the paper more exciting and uplifting, perhaps expanding on the profound implications of this research, its astrophysical significance. E.g., one could stress that this study opens up a new, exciting direction/frontier in relativistic plasma astrophysics, and perhaps speculate a little bit on how exactly it paves the way for future studies, what kinds of important collective plasma processes can be studied with this approach in the near future.*

- We thank reviewer for this comment and the final remarks of the paper have been improved to make it clear that this study opens up a completely unexplored frontier in relativistic plasma astrophysics.

- In addition, we have added an additional section in the Supplementary Information (Section 6), which suggests that future experiments may be able to achieve $N_{\text{D}} \sim N_{\text{S}}$ and $N_{\text{D}} \sim N_{\text{S}}$ into the several tens and hundreds by using thicker target materials.

I hope the Authors will find these comments useful and that addressing them will help improve this already very significant paper.

Dmitri Uzdensky

Reviewer #2 (Remarks to the Author):

The manuscript reports on the generation of an electron-positron pair beam using the 440 GeV/c proton beam from Super Proton Synchrotron at CERN, by using hadronic and electromagnetic cascades in graphite and tantalum targets, respectively. The total number of pairs and the fluence of the pair beam are measured experimentally at the location of the post-target profile monitor (10 cm downstream of the target exit), and the pair beam density is deduced assuming it has the same duration as the proton beam. The energy spectrum of the pairs is also measured, in the spectral range from 30 to 220 MeV, corresponding to about 2.5% of the pairs in the beam and thus representing mostly the energy tail of the pair beam and not the bulk of the pair beam. These experimental measurements are in very good agreement with FLUKA simulations modelling the hadronic and electromagnetic cascades. FLUKA simulations are then used to extrapolate the experimental measurements at the location of the target exit, and to provide critical physical quantities to assess the plasma nature of the pair beam, such as the Lorentz factor of the transformation from the laboratory to the zero-momentum frame ($\Gamma_0 = 4.5$) and the pair beam temperature (4 MeV). Using these numerical quantities, it is then concluded that the pair beam can be considered a plasma as its size is larger than the Debye length.

This work certainly represents an important step in an experimental project aiming at the study of collective plasma behaviours with pair beams, but its scientific impact is limited as the plasma nature of the pair beam is not demonstrated experimentally. No plasma collective effect is reported in the manuscript, which would represent the most convincing demonstration that the pair beam is indeed a plasma. The experimental measurement of pair density is not used, instead it is the extrapolated value from the FLUKA simulation, and Γ_0 and the pair temperature are entirely coming from the simulations. For example, in the Supplementary information the authors argue that the bulk of the pair beam in the longitudinal momentum distribution is the most important when assessing the collective behaviour of a plasma (Fig. 4 in SI shows that the most important particles are in the range 0-40 MeV), and not the extended spectral tail. Yet the experimental measurement misses the bulk and only characterise 2.5% of the pairs in the spectral tail (30-220 MeV). The main conclusion of the manuscript, namely the plasma nature of the generated pair beam, is thus not substantiated by the experimental measurements but mostly by the FLUKA simulations.

The work deserves to be published with the improvements suggested below, but given its limited scientific impact it is more appropriate for a more specialised journal, and I therefore do not recommend its publication in Nature Communications.

- We thank the reviewer for these comments and we would like the opportunity to respond to these directly. In particular, we would like to clarify that the spectral range sampled by the electron-positron spectrometer (30-220 MeV) actually corresponds to 40% of the pair spectrum, with just 10% corresponding to lower energy pairs (<30 MeV). Therefore, the measured spectrum corresponds to a significant proportion of the bulk, as opposed to 2.5% of the pairs in the extended spectral tail. We believe this confusion has arisen because it was not made clear enough in the original manuscript that it is not practically possible to measure the entire pair spectrum with a reasonable energy resolution (due to the finite divergence of the beam). Therefore we have clarified in the revised manuscript that a practical design was chosen to sample a central section of the pair beam cross-section (corresponding to approximately 10% of the total beam). This ensured that a significant proportion of the bulk spectra could be measured with a reasonable energy resolution.

- In addition, to make the argument stronger that it is reasonable to deduce the pair beam density assuming it has the same duration as the proton beam, an additional section has been added to the Supplementary Information (Section 2) that shows calculations of particle straggling of the beam being <5 ps (much less than the total beam duration with 1-sigma 250 ps).
- We have made it clearer that the role of the FLUKA simulations is to determine the pair characteristics that are not possible to measure directly, stressing that the simulations are validated against all the available experimental data.
- Finally, if the pair density measured 10 cm downstream of the target is used instead of the peak pair density corresponding to the immediate target rear (obtained from the FLUKA simulation), it does not change the conclusion that $N_{\text{pm}}/N_{\text{D}}$ and $N_{\text{pm}}/N_{\text{s}}$ are greater than 1 (according to the scalings provided in equations 1 and 2), and so the impact of the paper is not dependent on this extrapolation.

List of comments:

1) Lines 37-38 in the abstract. The sentence “The produced pair beams have a volume that fills multiple Debye spheres and are thus able to sustain collective plasma oscillations” needs to be reformulated to clearly state what data supports this conclusion, and what comes from experiment and what comes from simulation. This is critical for the soundness of the paper: it must be stated clearly how the conclusion is substantiated by the data.

- We thank the reviewer for this comment and we have made it clearer in the abstract the precise result that the size of the produced pair beams exceeds the characteristic plasma scales such as the Debye length and the collisionless skin depth. In addition, it has been made clearer in the main text that the transverse beam profile and the e^{\pm} energy spectra measurement are the two experimental measurements used to substantiate the conclusion of the paper.

2) Lines 185-186 in the conclusion. “the electron-positron beam we produced at HiRadMat can safely be assumed to behave as a relativistic pair plasma”: same comment as point 1), clearly state how this conclusion is substantiated by the data, and the type of data (experiment versus simulation) used to support this conclusion.

- To support the response to the previous comment, the data validating the predictive capability of the simulations has been clearly separated from the discussion of plasma characteristic length scales to avoid any confusion.

3) Line 144. “FLUKA simulations predict that without the magnetic field, low energy ($\lesssim 10$ MeV) scattered e^{\pm} can irradiate the screens, but when the electromagnet is activated, these lower-momentum pairs are swept away and the measured signals more closely resemble the true spectra.” Without magnetic field, the device is not measuring spectra at all (we can’t say that magnetic field improves and make the measurement resembles more closely the true spectrum, it is simply not a spectrum without B field). This confusing sentence needs to be reformulated, and to state explicitly how the spectrum is obtained with regards to a possible background subtraction. Do the authors subtract the magnet-off case from the magnet-on data, or oppositely do not subtract the magnet-off data because it is assumed that this background disappears when the magnet is turned on?

- We agree with this comment – it is indeed not strictly precise to refer to the measurement made without the electromagnet activated as a spectrum, and so this phrasing has been removed from the text. It has been replaced by an explanation of exactly how the spectra shown in Figure 3 are obtained: i.e. by subtracting the speckle background caused by high-energy radiation scattered around the experimental area. A more detailed discussion of this procedure has been added to the Supplementary Information (Section 4).

4) Lines 161-162. “We assess whether this density is sufficiently high for the pair beam to be considered a plasma”: the pair density is not the only important parameter for getting a plasma, as explained in the next

sentence and as shown in Fig. 4. Something like “We assess whether such a pair beam can be considered a plasma” is more appropriate.

- We agree with this comment, and this statement in the revised manuscript has been replaced by the statement that for collective behaviour to be observed the physical size of the plasma must exceed the Debye screening length and the collisionless plasma skin depth, before including a direct comparison of the beam dimensions with these length scales.

5) Lines 164-165. “if the number of particles per Debye screening volume is much greater than one”: the number of particles per Debye screening volume is N_D , and it's the total number of pairs N_{pairs} that needs to be much greater than N_D . This sentence should be corrected.

- We agree that this comment in it's original form is confusing, and so it has been replaced in the text by the more precise statement that $N_{pairs}/N_D > 1$.

Reviewer #3 (Remarks to the Author):

The authors have presented very exciting work, where, for the first time, enough electrons and positrons have been created simultaneously to open up the study of collective physics in electron-positron pair plasmas. Simulation results agree well with the experimental findings, assuring that the conclusions of the manuscript are well-founded. I think that the paper constitutes a breakthrough, and recommend publication enthusiastically.

- We thank reviewer for their very positive comments.

Please find attached the updated manuscript.

Regards,
Charles Arrowsmith (on the behalf of all the co-authors).

REVIEWERS' COMMENTS

Reviewer #1 (Remarks to the Author):

I have carefully read the revised version of the manuscript "Laboratory realization of relativistic pair-plasma beams" submitted by Arrowsmith et al. for publication in Nature Communications (Manuscript NCOMMS-23-63777A-Z). As I stated in my previous comments on the first version of the paper, my overall impression of this article is very positive. Furthermore, I feel that the Authors have satisfactorily addressed most of my comments. I am therefore happy to recommend the paper's acceptance for publication in Nature Communications.

At the same time, however, I would like to suggest a few optional comments, which I hope would help improve this paper even further but which I do not regard as being critical for acceptance:

- 1) The Authors are using the nonrelativistic expression for the collisionless plasma skin depth (which they call λ_s); but since the plasma is relativistically hot in the co-moving frame, I think it would be more appropriate to use the relativistic expression (in which effectively the particle rest-mass energy $m_e c^2$ is replaced by the average energy, i.e., $\langle \gamma \rangle m_e c^2$).
- 2) Following up on my previous Comment 9, what I had in mind is giving the actual values (e.g., in cm) of the key kinetic plasma scales, such as λ_D and λ_s , for reference.
- 3) I am still not satisfied with the Authors' response to my previous Comment 11. The Authors basically did the opposite to what I had suggested. Instead of ending the paper on a high note with enthusiastic, forward-looking take-away summarizing points about the general fundamental scientific significance of the paper and its broader astrophysical implications, they end it with some technical, albeit important, details of how this technique can be advanced in the future. I would suggest revising this last paragraph to make it more exciting.

I hope the Authors will find these suggestions useful.
However, once again, I recommend acceptance in any case.

Dmitri Uzdensky

Reviewer #2 (Remarks to the Author):

The revised manuscript is much improved, in particular by presenting explicitly the critical role of the FLUKA simulations for the interpretation of the experimental measurements in the main text. The authors also clarified the relative importance of low-energy and high-energy particles in the pair beam, giving a stronger weight to the experimental measurements in the conclusion. Overall, the soundness of the paper is greatly improved with a more balanced contribution from experiment and simulations, thus extending the scientific impact of the result with a considerable leap forward in our ability to experimentally generate the conditions of a relativistic pair plasma.

My comments were addressed except the first one related to the abstract: it needs to be extremely clear how the conclusion is reached, that is using both simulations (e.g. to estimate characteristic length scales) and experiments (e.g. for size and density). I suggest to replace the new sentence in red by something along these lines:

"We have carried out Monte Carlo simulations that agree very well with the experimental data and show that the characteristic scales necessary for collective plasma behaviour, such as the Debye length and the collisionless skin depth, are exceeded by the measured size of the produced pair beams."

Provided my comment on the abstract is addressed, I recommend publication of the revised manuscript in Nature Communications.

14th May 2024

Response to final reviewer comments for manuscript NCOMMS-23-63777A-Z

Dear Editor,

We sincerely thank the reviewers for their final comments. Our reply to each of their queries is detailed below. The changes to the manuscript have been highlighted using a red font in the revised manuscript.

Reviewer #1 (Remarks to the Author):

I have carefully read the revised version of the manuscript “Laboratory realization of relativistic pair-plasma beams” submitted by Arrowsmith et al. for publication in Nature Communications (Manuscript NCOMMS-23-63777A-Z). As I stated in my previous comments on the first version of the paper, my overall impression of this article is very positive. Furthermore, I feel that the Authors have satisfactorily addressed most of my comments. I am therefore happy to recommend the paper’s acceptance for publication in Nature Communications.

At the same time, however, I would like to suggest a few optional comments, which I hope would help improve this paper even further but which I do not regard as being critical for acceptance:

1) The Authors are using the nonrelativistic expression for the collisionless plasma skin depth (which they call λ_s); but since the plasma is relativistically hot in the co-moving frame, I think it would be more appropriate to use the relativistic expression (in which effectively the particle rest-mass energy $m_e c^2$ is replaced by the average energy, i.e., $\langle \gamma \rangle m_e c^2$).

2) Following up on my previous Comment 9, what I had in mind is giving the actual values (e.g., in cm) of the key kinetic plasma scales, such as λ_D and λ_s , for reference.

3) I am still not satisfied with the Authors’ response to my previous Comment 11. The Authors basically did the opposite to what I had suggested. Instead of ending the paper on a high note with enthusiastic, forward-looking take-away summarizing points about the general fundamental scientific significance of the paper and its broader astrophysical implications, they end it with some technical, albeit important, details of how this technique can be advanced in the future. I would suggest revising this last paragraph to make it more exciting.

*I hope the Authors will find these suggestions useful.
However, once again, I recommend acceptance in any case.*

Dmitri Uzdensky

- We thank the reviewer for these suggestions and we have rearranged the ordering of the last paragraph so that the technical discussion now comes before the final concluding statement.

Reviewer #2 (Remarks to the Author):

The revised manuscript is much improved, in particular by presenting explicitly the critical role of the FLUKA simulations for the interpretation of the experimental measurements in the main text. The authors also clarified the relative importance of low-energy and high-energy particles in the pair beam, giving a stronger weight to the experimental measurements in the conclusion. Overall, the soundness of the paper is greatly improved with a more balanced contribution from experiment and simulations, thus extending the scientific impact of the result with a considerable leap forward in our ability to experimentally generate the conditions of a relativistic pair plasma.

My comments were addressed except the first one related to the abstract: it needs to be extremely clear how the conclusion is reached, that is using both simulations (e.g. to estimate characteristic length scales) and experiments (e.g. for size and density). I suggest to replace the new sentence in red by something along these lines:

“We have carried out Monte Carlo simulations that agree very well with the experimental data and show that the characteristic scales necessary for collective plasma behaviour, such as the Debye length and the collisionless skin depth, are exceeded by the measured size of the produced pair beams.”

Provided my comment on the abstract is addressed, I recommend publication of the revised manuscript in Nature Communications.

- We thank the reviewer for this comment and we have modified the wording of the abstract as suggested.

Please find attached the updated manuscript.

Regards,
Charles Arrowsmith (on the behalf of all the co-authors).